# Effects of Inducible Nitric Oxide Synthase (iNOS) Gene Knockout on the Diversity, Composition, and Function of Gut Microbiota in Adult Zebrafish

**DOI:** 10.3390/biology13060372

**Published:** 2024-05-23

**Authors:** Yajuan Huang, Yadong Chen, Haisheng Xie, Yidong Feng, Songlin Chen, Baolong Bao

**Affiliations:** 1Key Laboratory of Exploration and Utilization of Aquatic Genetic Resources, Ministry of Education, National Demonstration Center for Experimental Fisheries Science Education, Shanghai Ocean University, Shanghai 201306, China; yjhuang230622@163.com (Y.H.); chenyd@ysfri.ac.cn (Y.C.); x1438755713@163.com (H.X.); fyd0531@126.com (Y.F.); 2National Key Laboratory of Mariculture Biobreeding and Sustainable Goods, Yellow Sea Fisheries Research Institute, Chinese Academy of Fishery Sciences, Qingdao 266071, China

**Keywords:** inducible nitric oxide synthase, zebrafish, gut microbiota, 16S rRNA, immune

## Abstract

**Simple Summary:**

The gut microbiome, a complex microbial community, is intricately linked to the host’s genetic background. The regulatory role of host genes on the microbiome has garnered considerable attention due to its impact not only on physiological and immune responses but also on the composition and functionality of intestinal microbes and the overall immune system. Nitric oxide synthase (NOS), through the production of nitric oxide (NO), participates in host defense, immune regulation, inflammatory responses, and autoimmune diseases. However, the effects of NOS on the composition and function of the zebrafish gut microbiota and the regulation of its homeostasis remain unclear. Genomic editing technologies have enabled the creation of zebrafish with specific genetic defects, making them a recognized model for studying host–microbiome–immune interactions. In this study, we characterized the impact of inducible nitric oxide synthase (iNOS) deficiency on the gut microbiota of zebrafish through 16S rRNA amplicon sequencing. Results revealed significant alterations in the microbial diversity and abundance in iNOS-deficient zebrafish, notably a reduction in *Vibrio* and an increase in *Aeromonas*. Transcriptomic sequencing of the gut confirmed functional changes, showing significant alterations in pathways related to the complement and coagulation cascades, PPAR signaling, cell adhesion molecules, Staphylococcus aureus infection, steroid synthesis, and bile acid synthesis. These pathways are crucial in pathogen clearance, inflammatory responses, and immune regulation, highlighting the significant role of the iNOS gene not only in microbial composition but also in gut immune and metabolic functions.

**Abstract:**

The gut microbiota constitutes a complex ecosystem that has an important impact on host health. In this study, genetically engineered zebrafish with inducible nitric oxide synthase (iNOS or NOS2) knockout were used as a model to investigate the effects of *nos2a*/*nos2b* gene single knockout and *nos2* gene double knockout on intestinal microbiome composition and function. Extensive 16S rRNA sequencing revealed substantial changes in microbial diversity and specific taxonomic abundances, yet it did not affect the functional structure of the intestinal tissues. Notably, iNOS-deficient zebrafish demonstrated a decrease in *Vibrio* species and an increase in *Aeromonas* species, with more pronounced effects observed in double knockouts. Further transcriptomic analysis of the gut in double iNOS knockout zebrafish indicated significant alterations in immune-related and metabolic pathways, including the complement and PPAR signaling pathways. These findings underscore the crucial interplay between host genetics and gut microbiota, indicating that iNOS plays a key role in modulating the gut microbial ecology, host immune system, and metabolic responses.

## 1. Introduction

Nitric oxide (NO) is an important biochemical signaling molecule that is involved in many important biological functions, such as regulating vasodilation, participating in nerve signaling, coordinating immune defense, and so on. Its synthesis is executed by a cadre of nitric oxide synthases (NOS), which includes the neural (nNOS), endothelial (eNOS), and inducible variants (iNOS). Whereas nNOS and eNOS are constitutively active and modulated by intracellular calcium flux, iNOS is activated in the presence of pro-inflammatory signals and has the capacity to generate copious quantities of NO [1,2]. Neuronal nitric oxide synthase is mainly involved in neurotransmission, it plays an important role in the clustering of zebrafish [3]. This inducible isoform, expressed predominantly within immune cells, assumes a critical role in antimicrobial actions and the modulation of inflammatory responses [4,5,6].

Within the context of gut integrity, NO’s contribution is observed through its regulation of mucosal perfusion, smooth muscle relaxation, and epithelial secretory functions, collectively influencing the gut microbiota’s composition and homeostatic balance [7,8]. Mammalian model research has elucidated that perturbations in NOS expression can precipitate marked alterations in gut microbial consortia, consequentially affecting immune reactivity and disease susceptibility, with conditions such as inflammatory bowel syndrome and adiposity being among the notable outcomes [9,10].

The development of fish gut microbiota is a complex process. Prior to hatching, the digestive tract of fish is not fully developed. Therefore, microbial colonization in the intestinal tract of fish only occurs post-hatching [11,12]. In zebrafish (*Danio rerio*), the colonization cycle is divided into four steps: (1) migration of environmental microorganisms to fish, (2) intestinal adaptation of these microorganisms, (3) migration of microorganisms from host to environment, and (4) environmental adaptation of microorganisms [13]. From sterile embryos to exposure to the aquatic environment, the microbiota of zebrafish begins to move from the water column into the gut and begins to be enriched, microbial abundance increases significantly from 4 to 8 days post-fertilization (DPF), and after the start of feeding, microorganisms are ingested from the ingested food and from the environment of the water column where they are present, undergo an acclimatization process in the gut, and then colonize the gut. At the same time, some of the microorganisms are excreted in the feces, achieving a dynamic balance [12,13]. Studies indicate that most gut microbiota are not randomly acquired from the environment but are influenced by the host and retained by various factors within the host [14].

The zebrafish gut microbiota, characterized by a core consortium predominantly comprising Proteobacteria, Firmicutes, and Bacteroidetes, mirrors the microbial composition evident in mammalian systems, suggesting that zebrafish-derived insights may hold translational relevance for human health [15,16]. Germ-free zebrafish studies have shed light on the profound influence of microbial colonization on the intestinal immune architecture’s development and operational capacity, accentuating the critical role of microbial constituents in immune system maturation [17,18]. Moreover, zebrafish have been employed to interrogate the impacts of genetic and environmental variables on gut microbiota, underscoring the significance of host genomic composition in the shaping of these microbial environments [19,20].

The regulatory role of iNOS in the governance of the gut microbiota is underscored by its involvement in the heightened production of NO during inflammatory states. Research in murine models demonstrates that iNOS-derived NO can modulate both the density and composition of the gut microbiota; however, the underlying mechanisms of this influence remain elusive [21]. It is postulated that NO may mitigate oxygen availability within the gut by engaging in reactions with molecular oxygen, thereby promoting the proliferation of anaerobic bacterial species while concurrently restraining the overpopulation of aerobic pathogens like *Escherichia coli* and *Salmonella* [22,23].

The gut microbiome, consisting of a multifarious consortium of microorganisms, is integral to the physiological and immunological operations in a broad spectrum of organisms, encompassing humans and established model species such as the zebrafish. A breadth of empirical studies has highlighted the pivotal influence of these microbial assemblages on the metabolism of nutrients, maturation of the immune apparatus, resistance to pathogens, and the maintenance of overall organismal health [24,25]. Illustratively, in murine models, aberrations in *PI3K* functionality have been linked to the disintegration of gastrointestinal microbial populations, potentially affecting the biosynthesis of select antimicrobials, cholic acids, and vitamin B1 [26]. Additionally, *SCT* gene ablation has been observed to alter both the constituency and numerical prevalence of the murine gut microbiota [27]. To this point, the majority of microbiome research has been preferentially focused on mammalian subjects, with rodents being a primary focus. However, in invertebrates, there is a growing compendium of evidence that suggests the midgut functions as an immunological organ, eliciting immune-responsive signaling cascades or the expression of various immune effector molecules in response to pathogenic incursions and modifications within the gut microbiome [28,29]. The symbiotic dynamics between the gastrointestinal microbiome and the host’s immune defenses have been extensively investigated. In piscine models, contrarily, the interactive mechanisms between host and microbial entities are less well defined [30]. Investigations into the piscine gut microbiome could substantially enhance the welfare of fish as well as the methodologies employed in aquaculture [31]. Due to their physiological and genetic congruence with mammals, zebrafish are positioned as an exemplary vertebrate model for probing the nuances of host–microbiota interplay, a status further bolstered by their transparent embryonic stage and a thoroughly delineated immune system [15,19].

The correlation between iNOS, NO, and the gastrointestinal microbiota in zebrafish remains underexplored. This lacuna presents a research avenue to elucidate the consequences of genetic alterations, such as iNOS gene disruption, on the diversity and functionality of gut microbiota. Given the zebrafish’s susceptibility to genetic manipulation and its status as a pertinent model organism, this line of inquiry is poised to shed light on the intricacies of immune modulation and microbial equilibrium [19].

## 2. Materials and Methods

### 2.1. Animals

In zebrafish genome, there are two iNOS genes, *nos2a* and *nos2b*. The construction of zebrafish with *nos2a* and *nos2b* single knockouts at the early stage has already been completed [32,33]. Crossbreeding of homozygous iNOS-deficient zebrafish gave rise to the F0 progeny that were heterozygous double mutants. Through serial inbreeding of the F0 cohort and subsequent genetic analysis, a number of iNOS-deficient homozygous zebrafish of both male and female were identified within the F1 population. These F1 zebrafish were then interbred to establish a consistent genetic lineage. Both the wild-type and the iNOS-deficient mutant strains were propagated and managed under identical environmental conditions. The aquaria system was recirculating, maintaining a photoperiod of 14 h of light to 10 h of darkness, with the water temperature regulated at 28 °C. Feeding occurred bi-daily, with a diet comprising live *Artemia salina* and formulated pellet feed. All experimental protocols involving animals were sanctioned by the Shanghai Ocean University’s Institutional Animal Care and Use Committee, aligning with the ethical standards for laboratory animal welfare. The relevant certificate number is SHOU-DW-2019-054.

### 2.2. Gut Microbiota DNA Extraction and 16S rRNA Amplicon Sequencing

Gut microbiota composition was assessed by collecting intestinal contents from euthanized fish under sterile conditions. Five-month-old adult zebrafish with average body length (3 cm ± 0.5) and average body weight (6 g ± 0.5) were selected as the research objects. Each group was set up with 3 biological replicates, and each biological replicate was mixed with 5 individual samples. DNA was extracted using the QIAamp DNA Stool Mini Kit (Qiagen, Hilden, Germany), following the manufacturer’s instructions. The V3–V4 regions of the 16S rRNA gene were amplified using primers 341F and 805R with the addition of Illumina adapter sequences 343F:TACGGRAGGCAGCAG; 798R:AGGGTATCTAATCCT. PCR conditions were as follows: initial denaturation at 95 °C for 3 min, followed by 25 cycles of 95 °C for 30 s, 55 °C for 30 s, and 72 °C for 30 s, with a final extension at 72 °C for 5 min. The PCR products were purified using the QIAquick PCR Purification Kit (Qiagen) and quantified on a Qubit 3.0 Fluorometer using the dsDNA HS Assay Kit (Thermo Fisher Scientific, Waltham, MA, USA). Sequencing was performed on the Illumina MiSeq platform, generating 250 bp paired-end reads.

### 2.3. Data Analysis

The initial preprocessing of sequencing data involved the removal and exclusion of reads characterized by either correct barcodes or primers, or those containing more than one ambiguous nucleotide, to ensure the integrity of downstream analyses. Subsequently, a stringent criterion of 99% sequence identity was applied to define the amplicon sequence variants (ASVs). Alignment of representative sequences from each ASV cluster was conducted within the QIIME2 platform [34]. Phylogenetic inference was performed within QIIME 2 using the fast tree algorithm. Further analyses encompassed the generation of rarefaction curves; the assessment of alpha diversity metrics (Simpson and Shannon index) was facilitated by QIIME 2. Beta-diversity, delineating dissimilarities in microbial community composition among samples, was assessed utilizing phylogenetic metrics and specifically weighted and unweighted UniFrac distances. Assessment of community similarity was undertaken through UniFrac analysis, incorporating weighted and unweighted principal coordinate analyses (PCoA). Subsequently, identical statistical methodologies, namely the Wilcoxon test for pairwise comparisons and the Kruskal–Wallis test for multiple group comparisons, were applied to discern specific microbial taxa exhibiting divergent abundances among the various groups. The heatmap, which reflects the similarity and dissimilarity of community compositions across different groupings (or samples) at various taxonomic levels, was performed with R 3.3.1 and Python 2.7 by analyzing the correlation coefficients between the top 20 dominant microbiota and different expression genes to identify microbiota that exhibit significant correlations with environmental variables.

### 2.4. Measurement of Nitric Oxide Content in Zebrafish Intestine

Nitric oxide (NO), which readily dissolves in water to form nitrite, was quantified using the Griess reagent colorimetric method to measure the concentration of nitrites in the intestinal tissues [35]. Three zebrafish intestines were weighed and homogenized in a glass homogenizer containing nitric oxide testing reagent (Addison, Suzhou, China). After thorough homogenization on ice, the mixture was centrifuged at 12,000 rpm for 10 min at 4 °C, and the supernatant was collected. Nitrite concentrations in the supernatant were determined using a nitric oxide assay kit (Addison). A standard curve for NO was constructed using sodium nitrite (NaNO_2_) according to the manufacturer’s instructions.

### 2.5. Intestinal Pathological Tissue Preparation

Zebrafish were subjected to a fasting period of three days, followed by anesthesia with 0.02% MS-222. The intestines were surgically removed using sterile dissection techniques, washed with PBS, and fixed in 4% paraformaldehyde (PFA) for 24 h. Subsequent steps included a gradient dehydration in ethanol, clarification in xylene, and embedding in paraffin wax. Intestinal sections were cut transversely to approximately 6 µm thickness using a Leica microtome. The sections were mounted on glass slides, stained with hematoxylin and eosin (H and E), and sealed with a neutral resin. The histological structure of the intestinal cross-sections was examined under a microscope.

### 2.6. Transcriptomic Analysis

Total intestinal RNA was extracted with a trizol reagent (Invitrogen, Waltham, MA, USA), and RNA quality and integrity were analyzed with a microspectrophotometer (Thermo Field). Intestinal transcriptome sequencing of extracted RNA using Illumina technology (Ouyi Biology, Shanghai, China). Sequencing was conducted on the Illumina HiSeq 2500 system, generating 100 bp paired-end reads. Data were processed and analyzed using the pipeline involving STAR for alignment and DESeq2 for differential expression analysis. Differential gene expression analysis using RNA-Seq data involves several critical steps to ensure accurate identification of differentially expressed genes. Firstly, raw sequencing reads are quality checked and trimmed using tools such as FastQC (Version 3) and Trimmomatic (Version 0.40) to remove adapters and low-quality bases. These cleaned reads are then aligned to a reference genome using a splicing-aware aligner like STAR or HISAT2, which accounts for RNA splicing events. Post-alignment, the aligned reads are assembled into transcripts and quantified using software such as StringTie (V 1.3.3.b) or Cufflinks (V0.17). This step involves normalization procedures to adjust for sequencing depth and transcript length, providing a set of expression values typically measured as fragments per kilobase of transcripts per million (TPM). Subsequently, differential expression analysis is conducted using statistical packages such as DESeq2 (V1.44) or edgeR (V4.2). These tools utilize models to account for biological variability and compute differential expression based on normalized count data. The results typically include statistics like log fold changes and *p*-values, adjusted for multiple testing errors using methods like the Benjamini–Hochberg procedure. The significant differentially expressed genes are further analyzed for biological interpretation through pathway analysis and functional enrichment using databases such as Gene Ontology (GO) and Kyoto Encyclopedia of Genes and Genomes (KEGG). This holistic approach provides insights into the molecular mechanisms underlying the biological conditions studied.

### 2.7. Real-Time Quantitative PCR (RT-qPCR) Analysis of Immune Gene Expression

Total RNA was extracted from tissues using the Trizol reagent (Invitrogen) and cDNA was synthesized using a reverse transcription kit (Novozan, Nanjing, China). Gene sequences were downloaded from the NCBI database, and primers were designed using SnapGene software (V7.2) (Table 1). The primers were subsequently synthesized by Sangon Biotech, Shanghai. The expression levels of relevant immune genes (*cfd*, *ier2a*, *rgs13*, *CR854897.1*, *cd36*, *ikbke*, *cd28*, *adgrf6*, *trafd*, *cd109*, *c3a.1*, *cfb*, *igsf9ba*, *c8g*, *c6*, *il26*, *il13*, and *c5*) were quantified by RT-qPCR. Gene expression levels were normalized to the *β-actin* gene expression using the 2^ΔΔCt^ method. All data were compared to the relative mRNA expression levels in the wild-type (wt) control group.

## 3. Results

### 3.1. Generation of iNOS-Deficient Zebrafish Mode

Through successive inbreeding of heterozygous iNOS deletion zebrafish and genomic analysis of their progeny by DNA sequencing (performed by Sangon Biotech, Shanghai, China) and clustal omega alignment, we successfully established a stable *nos2* deletion zebrafish model. In wild-type zebrafish, *nos2a* and *nos2b* encode 1081 and 1078 amino acids, respectively, and are composed of four functional domains. Results after gene editing showed that frame shift mutations occurred in *nos2a* and *nos2b* due to 5 bp and 4 bp deletion, respectively (Figure 1A). The encoded amino acids were prematurely terminated at the 84th and 326th amino acids respectively, resulting in large changes in the protein structure and affecting the normal function of its genes (Figure 1B). However, after deletion of the gene, the mutant showed a normal growth state and reproduction pattern. (Figure 1C).

### 3.2. Intestinal Nitric Oxide Levels in Zebrafish with Modified iNOS Expression

Quantitative assessment of intestinal nitric oxide (NO) concentrations showed a *nos2* dose-dependent decrease in NO levels, with iNOS null mutants exhibiting the lowest NO production compared to their wild-type counterparts. These findings, highlighted in Figure 1D, underscore the critical role of iNOS in intestinal NO synthesis, with pronounced effects observed in the *nos2^−/−^*-null zebrafish.

### 3.3. Histological Analysis of iNOS-Deficient Zebrafish Intestinal Tissue

Histopathological evaluation of intestinal samples from iNOS deletion zebrafish did not demonstrate significant structural deviations from wild-type controls. This suggests that iNOS gene ablation does not detrimentally influence the structural integrity or homeostasis of zebrafish intestinal tissue, as depicted in Figure 1E.

### 3.4. Comparative Gut Microbiome Diversity in Zebrafish with Variable iNOS Genotypes

In zebrafish lacking iNOS, gut microbial diversity and composition were not substantially affected as a whole. The 16S rRNA sequencing highlighted significant changes in microbial populations, with α diversity indices, such as Shannon (Figure 2A) and Simpson (*p* < 0.01, Student’s *t* test), significantly decreased in iNOS double-deletion mutants compared with wild types, and little change in iNOS single-deletion mutants (Figure 2B). The overall richness and uniformity of microbial species in samples within and between groups were basically the same, with no significant difference, indicating that the experimental samples within each group were relatively stable (Figure 2C). PCA analysis showed differences in PC1 and PC2 community structure between wild type and *nos2* gene deletion zebrafish (Figure 2D).

Phylum-level analysis showed that the number of *Proteobacteria* accounted for 54% in wild-type individuals but decreased to 34% in *nos2*^−/−^ mutant individuals. In contrast, taxa such as *Fusobacteriota*, *Firmicutes*, *Actinobacteria*, and *Cyanobacteria* were significantly less abundant in *nos2*^−/−^ zebrafish, but no significant changes were observed in *nos2a*^−/−^ and *nos2b*^−/−^ monogene mutants (Figure 3A). The genus *Aeromonas*, usually a minor constituent in the zebrafish gut, surged in prevalence within *nos2*^−/−^ mutants, rising from 2% to 18% (*p* < 0.05, Fisher’s exact test). Additionally, the proportions of *Marinifilaceae* and *Pseudomonas* escalated in *nos2*^−/−^ zebrafish, whereas levels of *Muribaculaceae*, *Bacteroides*, *Plesiomonas*, *Vibrio*, and *Lachnospiraceae* diminished (Figure 3B).

Differential microbial taxa presence among samples, as determined by the Kruskal–Wallis test, revealed disparities in 15 taxa (*p* < 0.05). Wild-type zebrafish harbored higher quantities of *Vibrio* and *Vibrionaceae*, in contrast with genetically modified counterparts (*p* = 0.01325). *Aeromonas* displayed variable proportions, peaking in *nos2a*^−/−^ mutants (17%), followed by *nos2*^−/−^ (9%), and least in wild-type (2%) (Figure 3C). The heatmap analysis of the total microbial biomass indicated a higher occurrence of *Vibrio*, *Vibrionaceae*, and *Barnesiellaceae* in wild-type zebrafish in comparison to *nos2*^−/−^ mutants. Conversely, taxa such as *Gemmatimonadaceae* and *Sphingomonas* were more prevalent in wild-type fish. *Cetobacterium* was most abundant in the *nos2*^−/−^ zebrafish microbiota, as detailed in Figure 3D. Quantitative comparison of microbial sequencing data across all genotypes revealed *Cetobacterium* to be predominantly derived from *nos2*^−/−^ zebrafish (80%). Furthermore, over 50% of *Bacteroides* and *Vibrio*, as well as *Vibrionaceae*, originated from *nos2b*^−/−^ and wild-type fish, respectively. Strikingly, *nos2b*^−/−^ mutants contributed 95% of *Haemophilus*, while *nos2a*^−/−^ mutants were the main source of *Lactobacillus* and *Pseudomonas*, exceeding 50% (Figure 4).

### 3.5. Transcriptomic Insights from nos2^−/−^ Models

Transcriptome sequencing of gut tissues indicated the presence of 13,269 common genes between both experimental cohorts. Unique to the *nos2*^−/−^ group were 1408 genes, whereas the wild-type group exhibited 953 distinct genes (Figure 5A). Comparative analysis disclosed 3305 genes with differential expression, comprising 1790 upregulated and 1515 downregulated genes in the *nos2*^−/−^ zebrafish; some of the genes with significant changes were also flagged (Figure 5B). GO enrichment analysis of the upregulated genes demonstrated a preponderance in immunological functions, notably in complement activation, lectin pathway, elastic fiber assembly, defense response to Gram-negative bacteria, response to estradiol, immune effector process, and other aspects. (Figure 5C).

KEGG pathway enrichment analysis elucidated that *nos2*^−/−^ zebrafish exhibited a significant participation of upregulated genes in the Complement and coagulation cascades pathway, hinting at an intensified inflammatory response. Additionally, the drug metabolism–cytochrome P450, PPAR signaling pathway, retinol metabolism, and porphyrin metabolism were found to be upregulated (Figure 5D).

After functional annotation screening of the transcriptome, a total of 37 genes were identified with differential expression between gene-edited individuals and wild-type individuals. Among these, five genes were downregulated: *btg2* (B-cell translocation gene 2), *cd109* (CD109 molecule), *hs2st1b* (heparan sulfate 2-O-sulfotransferase 1b), *nfil3-6* (nuclear factor, interleukin 3 regulated, member 6), and *hsp70.2* (heat shock cognate 70-kd protein). Additionally, 31 genes were found to be upregulated: *emc2* (ER membrane protein complex subunit 2), *itln3* (intelectin 3), *ambp* (alpha-1-microglobulin/bikunin precursor), *apoa2* (apolipoprotein A-II), *saa* (serum amyloid A), *apoda.2* (apolipoprotein Da, duplicate 2), *ccl39a.10*, chemokine (C-C motif), *ligand 39* (duplicate 10), *thrsp* (thyroid hormone responsive), etc., as well as seven complement regulatory factors such as CFI, CFH, and CFB, and 15 components of the complement system including C1, C3, C5, C7, C8, etc. (Figure 6A). After iNOS gene editing, a subset of downregulated genes was concentrated in certain cell surface factors, B-cell regulatory factors, interleukin regulatory factors, and heat shock proteins. Following the knockout of iNOS, a number of chemotactic factors were upregulated, and most complement genes were also upregulated, with the fold increase ranging from 2 to 260 times. These differentially expressed genes may have potential roles in relation to microbial diversity.

### 3.6. Predictive Analysis of Correlations between Gut Microbiota and Gene Expression Profiles

Notable correlations were elucidated between gut microbial composition and host gene expression alterations. The changes in the gut microbiota have a certain relationship with the expression characteristics of genes (Figure 6B). There is a significant positive correlation between *Cetobacterium* and multiple complement genes. In addition, *Aeromonas* has a significant positive correlation with the CFH gene, but a significant negative correlation with the *btg2* gene and the *nfil3* gene. Furthermore, *Vibrio* bacteria, which are significantly downregulated in *nos2*^−/−^ mutant individuals, show a significant negative correlation with several complement genes. On the contrary, *Enterobacteriaceae* have a significant positive correlation with the majority of complement genes. Species such as *Gemmatimonadaceae*, *Sphingomonas*, *Escherichia-Shigella*, *Corynebacterium*, *Lachnospiraceae NK4A136* group, *Shinella*, and *Leptotrichia* exhibit similar characteristics, showing a significant negative correlation with multiple complement genes, among which the abundance change characteristics of *Escherichia-Shigella* are highly significantly negatively correlated with complement genes.

### 3.7. Validation of Immune-Related Gene Expression Alterations

Quantitative PCR was employed to validate a subset of 18 genes, revealing pronounced immune-related differential expression. The analysis confirmed substantial downregulation of genes such as *cfd*, *ier2a*, *rgs13*, *CR854897.1*, *cd36*, *ikbke*, *cd28*, *adgrf6*, *trafd*, and *cd109*, while *c3a.1*, *cfb*, *igsf9ba*, *c8g*, *c6*, *il26*, *il13*, and *c5* were markedly upregulated, aligning with the transcriptomic observations (Figure 6C,D).

## 4. Discussion

The observed changes in the gut microbiota of iNOS-deficient zebrafish, especially the decline in α diversity and changes in microbial community, highlight the important role of nitric oxide (NO) in gut microbiota regulation. We noted a decrease in proteobacteria and firmicutes, consistent with the effect of iNOS on microbial composition in mammalian models under inflammatory stress [36]. The marked increase in *Aeromonas* spp. in these zebrafish points to a microbial adaptation to diminished NO levels, as *Aeromonas* proliferates when host antimicrobial mechanisms are weakened [37].

The interaction between NO and microbial dynamics is partially attributable to NO’s function in mucosal immunity and the management of oxidative stress. NO is known to modulate gut oxygen levels, favoring anaerobic bacteria growth while inhibiting aerobic bacteria proliferation [38,39]. This may explain the rise in *Fusobacteriota*, an anaerobic group, in the absence of iNOS. Additionally, the upregulation of inflammatory markers in our transcriptomic analysis signifies that iNOS deficiency alters the immune environment, potentially impacting the gut’s ability to manage its microbiota [10]. Previous studies have shown that *Aeromonas* presence can significantly upregulate the host iNOS gene expression [40], and iNOS knockout zebrafish have displayed an increased *Aeromonas* presence, reinforcing NO’s role in the intestinal barrier and the antimicrobial function [41]. This alteration could increase infection risks, emphasizing NO’s crucial role in gut pathogen control. Conversely, a decrease in Vibrio, which is normally controlled by NO-mediated immune pathways, indicates a disruption in microbial regulation [42].

The complement system is pivotal in immune surveillance, and its roles extend to maintaining host health and combating pathogens. It aids in clearing debris, coordinating immune responses, and signaling danger, thus maintaining internal balance [43]. The absence of the iNOS gene leads to significant complement pathway activation, indicating that iNOS loss triggers a strong immune reaction and is vital in the immune cascade. The overexpression of genes in the PPAR signaling pathway in iNOS knockout fish is consistent with PPAR’s role in microbial infection response and inflammation management [44]. PPAR, a principal transcription factor, modulates various immune and inflammatory genes and affects intestinal metabolism via the gut microbiota [45]. Suggesting a compensatory mechanism for the altered microbial load or composition.

In humans, P450 enzymes are essential for xenobiotic defense, facilitating the metabolism of external substances and converting arachidonic acid into bioactive molecules [46]. They play a key role in detoxifying drugs, processing toxins, and synthesizing essential compounds like steroids and cholesterol [47] and are pharmacological targets for certain cancers and diseases [48]. The dramatic upregulation of the P450 metabolic pathway following iNOS gene deletion suggests that iNOS is not only involved in immune and inflammatory responses but may also influence drug metabolism, highlighting its potential as a target for drug development.

It has been shown that iNOS is a metabolic enzyme involved in the regulation of a variety of metabolic functions in organisms, including glycolipid metabolism [49]. The observed inhibition of metabolic pathways, particularly those involved in the processing of lipids and carbohydrates, may be related to the deletion of the iNOS gene, or it may be a systemic physiological modulation of the altered composition of the intestinal microbiota. Metabolic activities are intricately linked to the microbial milieu, with microbiota fluctuations profoundly influencing the host’s metabolic functions [50]. This dynamic underscores the multifaceted nature of host–microbiome interactions, where shifts in the microbiome not only modulate immune functions but also reshape metabolic routes, thereby affecting disease vulnerability [51]. Subsequent correlation analyses reveal a robust network of associations between diverse microbial communities and host genomic expressions, aligning with findings from host gene–microbe interaction studies [52].

Our results augment the existing research, underscoring the utility of zebrafish as a pertinent model for gut microbiome–host studies. Analogous research in murine models has demonstrated that iNOS deficiency heightens colitis susceptibility, concomitant with gut microbiota alterations [53]. This parallelism indicates that pivotal processes in microbiota–immune system interactions are preserved among vertebrates, bolstering the validity of zebrafish for gastrointestinal research [54].

The findings herein pave the way for further inquiries. Investigations into the proliferation of specific microbial groups in the absence of iNOS could shed light on their contributions to health and pathology. Furthermore, analyzing the consequences of shifted metabolic pathways might reveal therapeutic targets for metabolic syndromes [55].

Clinically, discerning the role of iNOS in gut microbiota modulation could inform the creation of novel treatments focused on nitric oxide pathways. Such strategies may be particularly pertinent for conditions marked by microbial imbalance and inflammation, like inflammatory bowel disease and metabolic syndrome [56]. While our research offers valuable insights into iNOS’s role in gut microbiota regulation, it has its constraints. The direct health impacts of altered Aeromonas levels were not examined, necessitating future studies to ascertain the pathogenic implications of this microbial alteration. Moreover, translating our results to human health demands further corroboration through clinical trials [57].

## 5. Conclusions

In conclusion, the mutant model retained normal physiological function, but its nitric oxide synthesis was significantly reduced, and iNOS deletion led to significant changes in the gut microbiota found in the zebrafish. Despite the absence of significant histological changes in the intestinal tissue, the iNOS-deficient zebrafish displayed a profound reduction in intestinal nitric oxide levels and substantial shifts in gut microbial diversity, underlining the crucial role of iNOS in intestinal homeostasis and microbial regulation. Notably, the iNOS mutation led to a distinct microbial community structure, with significant changes in microbial taxa abundance and diversity, influencing the presence of specific microbial genera and leading to a surge in certain opportunistic pathogens. Transcriptomic profiling revealed a differential gene expression pattern, with the upregulation of genes involved in the inflammatory response and a correlation between changes in gut microbiota and host gene expression profiles, particularly those related to the complement system. These findings suggest that iNOS gene plays a pivotal role in modulating the gut microbiota through host–microbial interactions, impacting both innate immunity and the inflammatory response in zebrafish.

Of course, the composition of gut microorganisms is affected by a variety of complex factors, such as the environment, individual genetic background, and other factors. Although we have tried to minimize the influence of complex factors, our results can only provide a reference for the relevant research and do not guarantee its absolute validity; more effective research requires more investigation.

## Figures and Tables

**Figure 1 biology-13-00372-f001:**
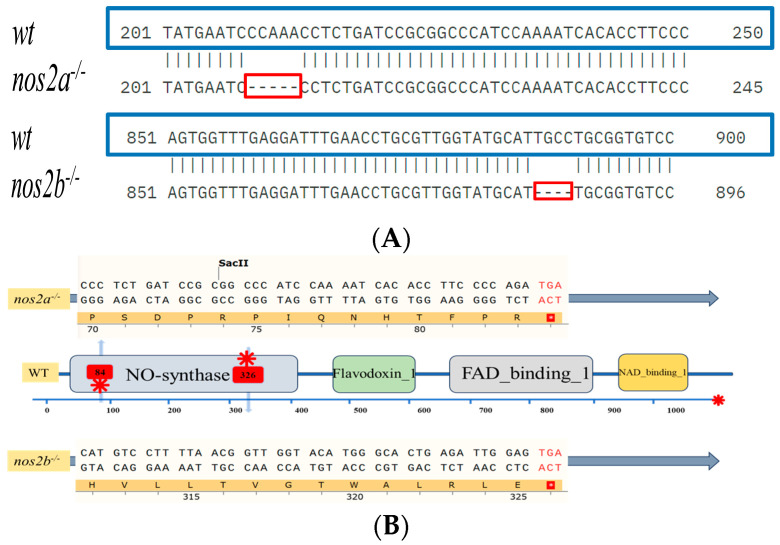
Characterization of zebrafish genotypes and intestinal analyses. (**A**) Results of zebrafish gene knockout verification. (**B**) Amino acid coding results of *nos2a* and *nos2b* gene knockout in zebrafish (**C**) Images of zebrafish with different genotypes. (**D**) Measurement of nitric oxide (NO) levels in the intestines of different genotypes, **** *p* < 0.0001. (**E**) Histological sections of intestinal tissues from zebrafish with various genotypes (5 months old adult zebrafish).

**Figure 2 biology-13-00372-f002:**
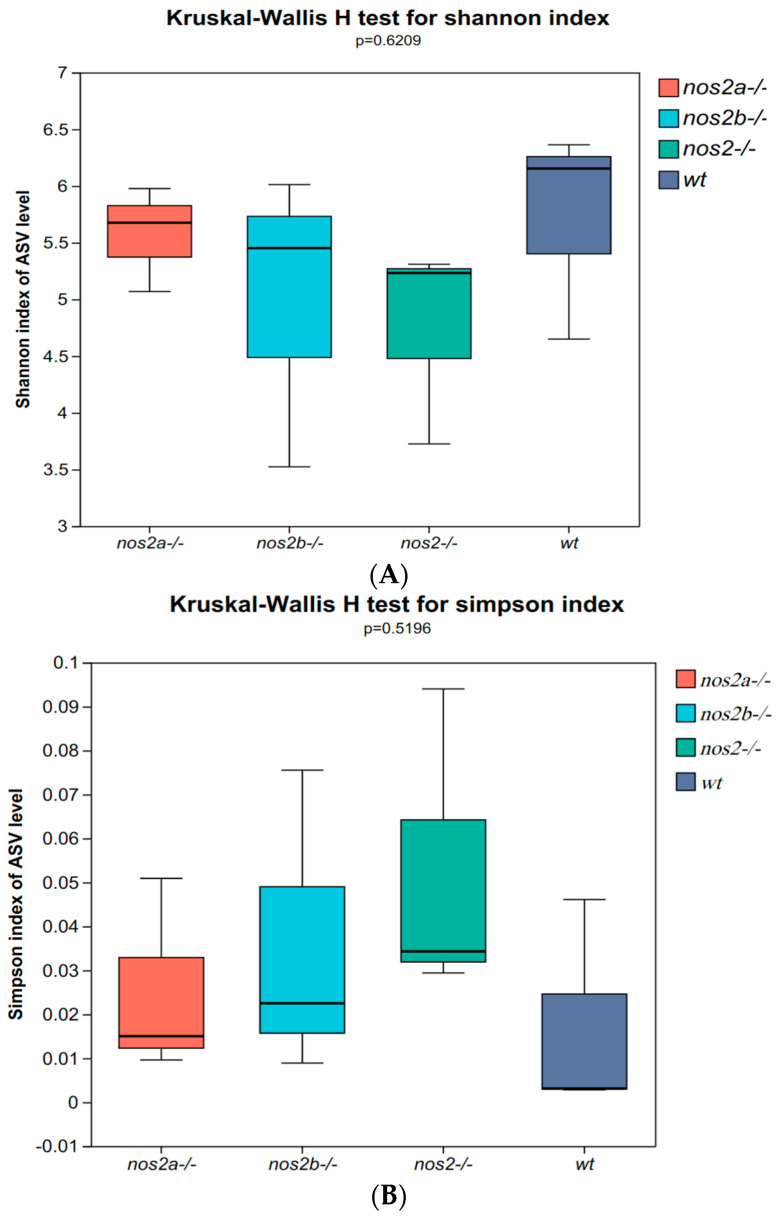
Diversity and principal component analysis of microbial communities. (**A**) Shannon index calculated using the Kruskal–Wallis H test; larger Shannon values indicate higher community diversity. (**B**) Simpson index calculated using the Kruskal–Wallis H test; the greater the value of Simpson’s index, the lower the diversity of the community. (**C**) Richness and evenness of species within the samples: Species richness is indicated by the width of the curve along the horizontal axis—the wider the curve (greater horizontal spread), the more diverse the species composition. The evenness of the species composition is reflected by the shape of the curve—the flatter the curve (smaller vertical spread), the higher the evenness. (**D**) Principal Component Analysis (PCA): By decomposing variances, PCA displays the differences among multiple datasets on a coordinate plot. Axes represent the principal components with the largest variance. The closer the clustering of points within the same group, the better the stability within that group. The greater the distance between different groups, the larger the differences between them.

**Figure 3 biology-13-00372-f003:**
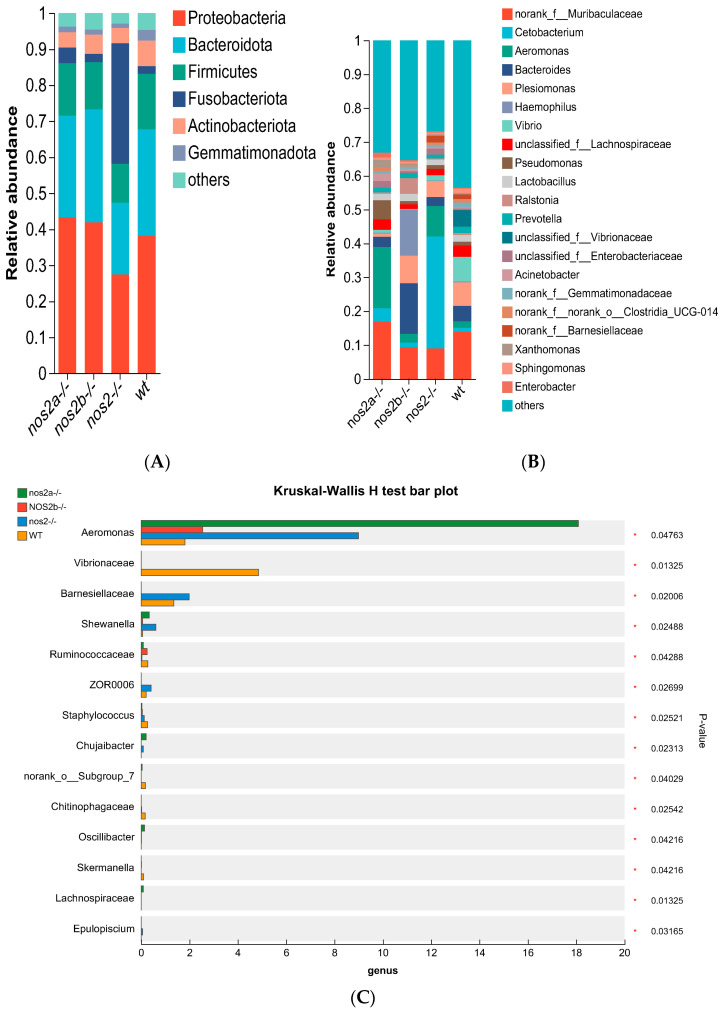
Analysis of gut microbiota composition at different taxonomic levels. Changes in the microbial community at the phylum level between wild-type and iNOS knockout zebrafish. Bar charts represent the mean ± SEM, *n* = 3 per group * *p* < 0.05. (**A**) Phylum-level analysis of different samples. (**B**) Composition of gut microbiota at the genus level in different samples. (**C**) Differences in microbial communities at the genus level among different samples, analyzed using the Kruskal–Wallis algorithm. (**D**) Heatmap analysis of the total microbial biomass.

**Figure 4 biology-13-00372-f004:**
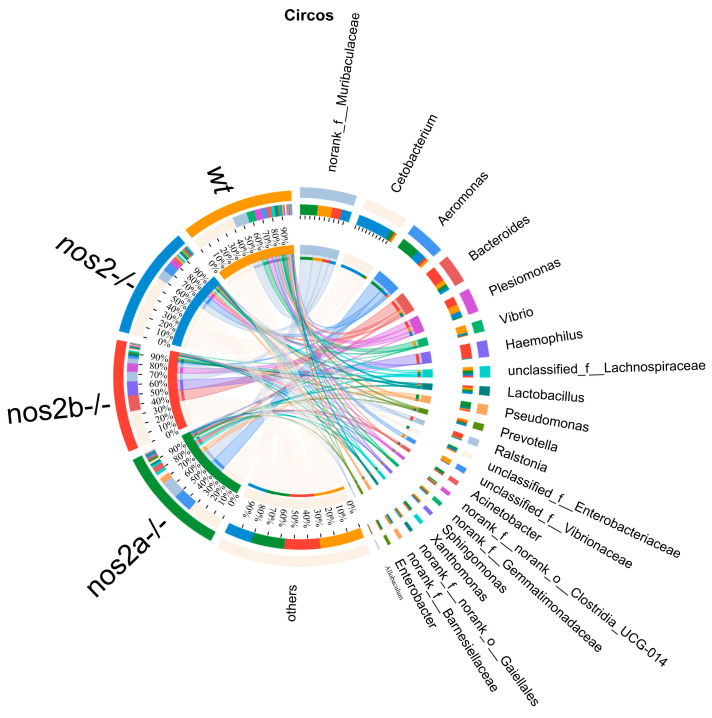
The Circos plot illustrates the relationship between microbial abundance and host genotypes. The outermost ring of the plot depicts segments corresponding to various bacterial genera, with their names duly annotated. Inner bands connect these segments to a range of host genotypes, including wild-type (wt) and inducible nitric oxide synthase (iNOS) knockout, indicating the proportion of each bacterial genus present in each genotype. Adjacent to the names of the bacteria, colored bars represent the relative abundance within the respective genotypes. The breadth and color intensity of the bands are indicative of the level of association, signifying the variation in bacterial distribution across different genotypes.

**Figure 5 biology-13-00372-f005:**
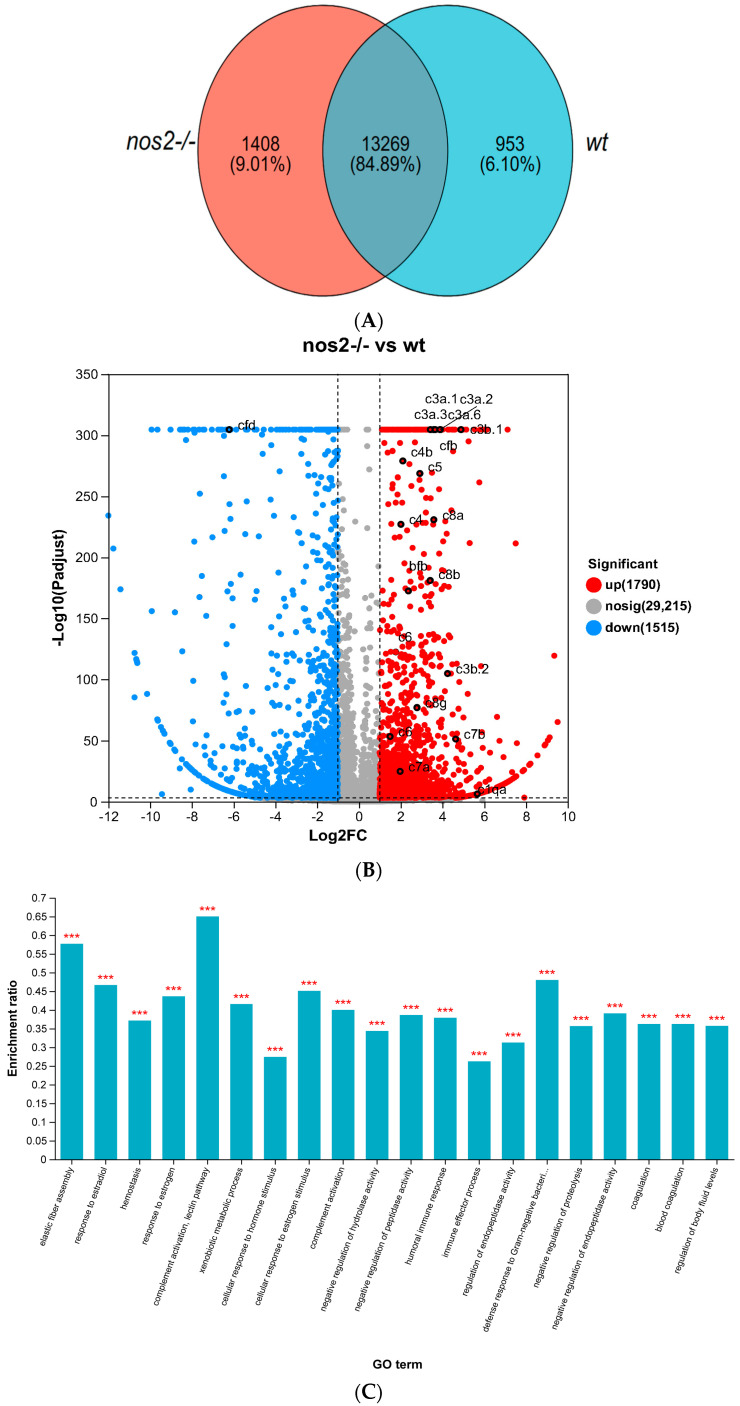
Transcriptomic Sequencing of *nos2*^−/−^ mutant and wild-type (WT) groups. Heatmap of differentially expressed genes associated with key metabolic and immune pathways. Expression levels are depicted as log2 fold change between knockout and wild-type groups. (**A**) Distribution of shared and unique genes between the *nos2*^−/−^ mutant and the wild-type group (WT). (**B**) Number of differentially expressed genes between the *nos2*^−/−^ mutant and the wild-type group (WT). (**C**) Gene ontology (GO) enrichment analysis, *** *p* <0.001. (**D**) Enrichment of pathways in the Kyoto Encyclopedia of Genes and Genomes (KEGG). All statistical analyses were conducted using R software version 3.6.1. Differences were considered significant at *p* < 0.05.

**Figure 6 biology-13-00372-f006:**
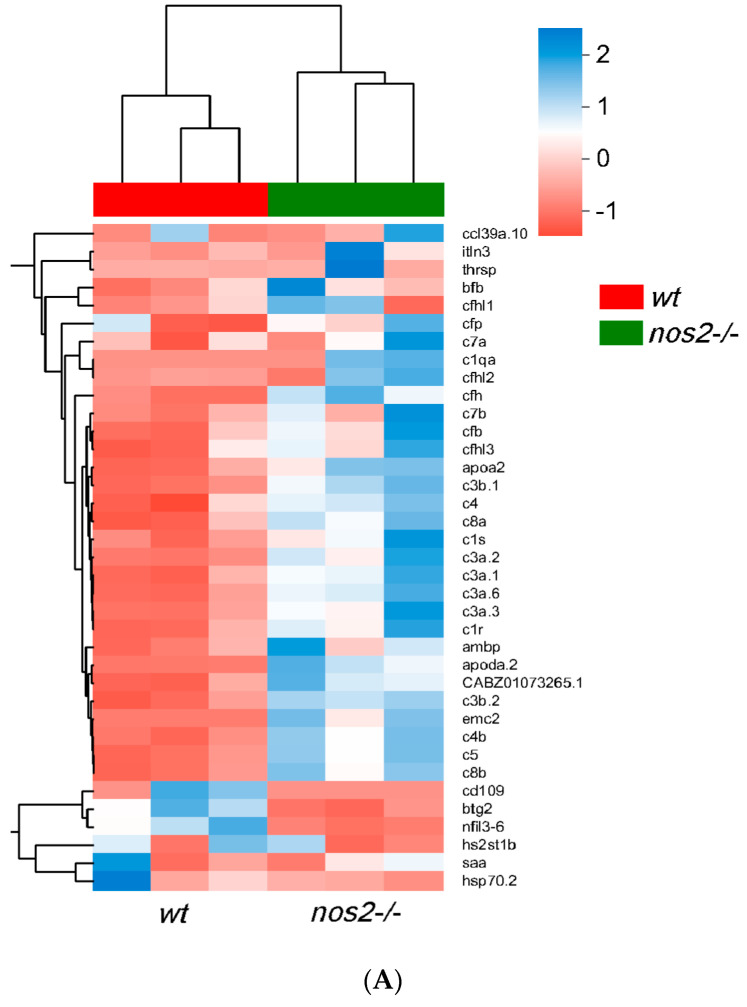
RT-qPCR validation of changes in mRNA expression levels of immune genes normalized to mRNA expression levels in wild-type (WT). (**A**) Significant differences in gene expression between the *nos2*^−/−^ mutant group and the wild-type (WT) group. (**B**) The horizontal axis is the gene name, the vertical axis is the microorganism; the asterisk indicates that the *p*-value is less than 0.05, which is represented by *; the color indicates the correlation, red indicates the positive correlation, and blue indicates the positive correlation. (**C**) mRNA expression levels of upregulated genes. (**D**) mRNA expression levels of downregulated genes.** Data represent three independent experiments. Note: Asterisks (*) denote statistical significance; ‘ns’ indicates no statistical significance; * *p* < 0.05, ** *p* < 0.01, *** *p* < 0.001, **** *p* < 0.0001.

**Table 1 biology-13-00372-t001:** Quantitative PCR primers.

Primer Name	Primer Sequence 5′-3′	Amplification Length
*cfd*-F	GACTGTATCACGGGAGGGC	>NM_001020532.1
*cfd*-R	CGGCTCCAGACAACGAGT	
*ier2a*-F	ATCTGTCACAGAGGAGAGCAC	>NM_001142583.3
*ier2a*-R	GGGCTGCAGTTTTCTTTGTCC	
*rgs13*-F	GAGAAGTGGCGGTTTCGTTG	>NM_001089512.1
*rgs13*-R	CTTGAAATCTTCGCAGGCCAG	
*card*-F	GCCATGAAGCCCTCAACAC	>XM_021466485.1
*card*-R	ACCTCATGCTCATGACTGATCT	
*CD26*-F	CGGAATCCTCATCCCGGTG	>NM_001002363.1
*CD26*-R	CTGCACCAACACAGGTTTGG	
*ikbke*-F	GTGGCTCATGGAATGAACCAC	>NM_001002751.2
*ikbke*-R	TCCGGATGCAGATATTCCTCTG	
*CD28*-F	CTGGGATTCGCTCTGGTTGT	>XM_005167519.4
*CD28*-R	GCGAAGTAGGGCTCTGAGAG	
*adgrf6*-F	TGTCTAAAGTCACCTGAGGTCTG	>XM_017352625.2
*adgrf6*-R	CCTCAGCTGATCAGTGATCCT	
*trafd1*-F	TGTGAAGTGTGCCAGGAGC	>NM_001089515.1
*trafd1*-R	AGGAGGCCAGCTCTAGATCA	
*CD109*-F	GTCTGAAGTTCTGCGCTGTG	>XM_692420.9
*CD109*-R	CTCCACATTCAGCTGATACGGA	
*c3a.1*-F	GTGTGACCCGCTATATGTGC	>NM_131242.1
*c3a.1*-R	AGAACAAGTTCTGATCATCAGG	
*cfb*-F	TCAATTTGGAGTCTGGACCCC	>NM_131338.2
*cfb*-R	GTTGGCAAACCCGGACTTTC	
*igsf9ba*-F	TGTGGAGTGGTTCAAATTCGG	>XM_009291702.3
*igsf9ba*-R	AAGGTATCGTACTGTTGCTCCAG	
*c8g*-F	TTCGCTTCTGGCTGTATTTGTT	>NM_200863.1
*c8g*-R	CCACTTGCCACTCATCTGATC	
*c6*-F	TCCCACTATGGGGTGTTTCTG	>XM_017352961.2
*c6*-R	GCAGGCCTCAACATTACACA	
*IL26*-F	CAGGAGGAATGTTTGAAGCGG	>NM_001020799.1
*IL26*-R	TCCAGGACACGCTTGAAGTC	
*IL13*-F	CTCGCCTGCACTGTATTCG	>NM_001199905.1
*IL13*-R	AATCATGCTCACACTTCAGGC	
*c5*-F	TGTCTGCTTCACCGTTCAGG	>XM_001919191.8
*c5*-R	GCGTTGAGCTTCAGGGATTG	

## Data Availability

The data are contained within the article.

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
