# Peer review of "Effects of Inducible Nitric Oxide Synthase (iNOS) Gene Knockout on the Diversity, Composition, and Function of Gut Microbiota in Adult Zebrafish"

_biology, 2024, doi:10.3390/biology13060372_

Round 1

Reviewer 1 Report

Comments and Suggestions for Authors

Comments:

I carefully evaluated the manuscript submitted by Yajuan Huang et al. entitled “Effects of inducible nitric oxide synthase (iNOS) gene knockout on the diversity, composition, and function of gut microbiota in adult zebrafish”.  The authors had done extensive research on how knockout of iNOS is affecting the gut microbiome and allied pathways in zebrafish. It can be accepted with minor revision to publish in MDPI Biology journal.

Ø  Authors mentioned that in Simple summary that “Transcriptomic sequencing of the gut confirmed functional changes, showing significant alterations in pathways related to the complement and coagulation cascades, PPAR signaling, cell adhesion molecules, Staphylococcus aureus infection, steroid synthesis, and bile acid synthesis”.

My question is, did authors perform Staphylococcus aureus infection studies after iNOS gene knockdown in zebrafish? Because no where in the text was mentioned except in simple summary.

Ø  In Figure 1B, the Wt and nos2-/- zebrafish are alike with protruding mouth and the other two (nos2a-/- & nos2b-/-) are different. Is there any link between iNOS knockdown and mouth morphology? Similarly, the fin patterns have differences. Is it just fixation issue or impact of iNOS gene knockdown? Authors has to explain.

Reviewer 2 Report

Comments and Suggestions for Authors

Reviewer 3 Report

Comments and Suggestions for Authors

The article titled “Effects of inducible nitric oxide synthase (iNOS) gene knock- out on the diversity, composition, and function of gut microbiota in adult zebrafish” explores the interaction between the host’s genes and the gut microbiota. The authors explore the role of Nitric oxide (NO) in maintaining the diversity and abundance of microbial communities in the gut given the myriad of functions that NO regulate in the gut. The authors therefore used a genetically engineered inducible Nitric oxide synthase (NOS) homozygous mutants to explore the role that NO plays in mediating the interactions between the host genes and gut microbiota.

General comments: Overall, the authors test the role of NO on the gut microbiota and gene expression. The relationship between the microbiota and intestinal gene expression is not strongly established except for a correlation analysis. The authors separately looked at changes in microbial diversity and gene expression in the gut separately. Is it possible to do the same experiment in Germ-free fish to see what genes are altered even in the absence of microbiota. My major concern in this manuscript is the presentation. The figures and figure legends could be improved significantly.

Major comments:

-              The figures in this manuscript can be made in a better way. Except for Figure 1, all other figures can be improved. Text sizes and panel orientation can be improved throughout. Very hard to read the text in figures. Why not have panels beside each other rather than below each other?

-              In Figures 2A and B, the p-values are shown are greater than 0.5. So, it’s not clear whether the differences observed from these statistical tests are significant or not? Also, I can’t interpret what the Y-axis means in Panels 2A and 2B. Same goes for Figures 2C and 2D – please refer to these panels in the text and elaborate the rationale behind these statistical tests and what were deduced.

-              Section 3.6, Lines 386-399: The title of this section is misleading. The authors do a correlation analysis between the bacterial species and genes that are expressed in the gut which is a computational model. The interplay can only be tested if specific microbial populations are removed from the gut and the gene expression assayed. Please change the title of this section to reflect that the analysis performed here is a computational prediction.

-              Lines 383 – 384: The authors observed changes in gene expression in iNOS mutants and also observed changes in microbial populations. But that does not suggest that the “changes in gene expression have an important impact on the colonization of microbes in the gut”. If the authors want to claim this, maybe they should manipulate these genes and assay for changes in microbial populations.

Minor comments:

-              Line 54–55: Not sure I understand what the authors mean by “clustering in zebrafish”? Do the authors mean neuronal clustering?

-              Line 6869: Does the microbiota enter the gut from the environment as the fish imbibes water into its gut? Maybe please add a sentence about that in this section.

-              Lines 133-134: Were the mutant animals maintained as homozygous mutants? Can the authors be explicit here? If so, did that end up selecting for a compensatory effect of other orthologous genes?

-              Line 239: Change “modle” to “model”

-              Figure 1: What stage is shown in Panel D? Please include that in the figure legends.

-              Also Figure 1A looks a bit distorted please fix that.

-              Figures 2C and 2D are not referred to in the text.

-              It’s also not very clear the point of Figues 2C and 2D. I suggest the authors spend a few sentences describing the rationale behind those panels and what those panels suggest.

-              Figure 5B: Can the authors label some of the top differentially expressed genes in this plot. Also talk about what the authors interpret from the altered expression of these genes.

-              In Figure 5C, what does the color bar mean? Why are there no high and low boundary points in the color bar legend?

-              Panel 6B is extremely hard to read. Please make the panel bigger.  

-              Line 344-353: Can the authors provide a table of the genes associated to each GO term and each KEGG pathway. Also, for these genes, can the authors show how the expression of these specific genes within each pathway were altered?

Comments on the Quality of English Language

English language is ok, with some improvements required. Some statements are quite leading which means that they indicate a causal relationship when such a relationship is not experimentally shown. Overall can be improved. Please explain each panel in some detail for each figure. 

Reviewer 4 Report

Comments and Suggestions for Authors

The paper entitled "Effects of inducible nitric oxide synthase (iNOS) gene knockout on the diversity, composition, and function of gut microbiota in adult zebrafish" has great potential and its consistent with the scope of the special issue. In addition to this, the paper needs some revision before final scientific discus. For that reason, I have prepared a suggestion and recommendation list to help. You can find the list in followings. Good luck.

Introduction 

The introduction provides a broad overview of nitric oxide (NO) and its synthesis but lacks detailed background on previous studies that directly link iNOS with gut microbiota in zebrafish. Including more specific previous research would strengthen the context.

M&M

Justify the sample sizes chosen for the experiments.

Results

Well

Discussion

Sometimes overgeneralizes the findings from zebrafish to other vertebrates, including humans, without sufficient justification. More cautious interpretation and a discussion on the limitations of extrapolating these results would be appropriate.

Lacks integration with a broader range of existing literature. Comparing and contrasting the findings with other studies on iNOS and gut microbiota in different models could provide a richer context.

That part does not address potential confounding factors that could have influenced the results, such as environmental variables or genetic background differences. Acknowledging and discussing these factors would strengthen the study's validity.

Comments on the Quality of English Language

Although the general language of the paper is good, I suggest you revise the sections listed below in terms of language and expression.

Line 31-33: The gut microbiome constitutes a complex ecosystem with a significant impact on host health. In this study, genetically engineered zebrafish with inducible nitric oxide synthase (iNOS) knockouts were used as models to investigate the effects of single and double homozygous knockouts on the composition and functionality of the intestinal microbiome

Line 47-49: Nitric oxide (NO), a paramount biochemical mediator, is implicated in a myriad of vital biological operations, including the modulation of vasodilation, signal transmission in neural structures, and the orchestration of immune defenses.

Line 195-199: Total RNA was extracted from the gut tissue using the TRIzol reagent (Invitrogen, USA) according to the manufacturer's protocol. The quality and integrity of RNA were evaluated using the Agilent 2100 Bioanalyzer (Agilent Technologies, USA). Libraries for RNA sequencing were prepared using the TruSeq Stranded mRNA LT Sample Prep Kit (Illumina, USA), following the manufacturer’s guideline.

Line 268-271: Phylum-level analysis revealed a decrease in Proteobacteria, from 54% in wild-type to 34% in nos2-/- mutants. In contrast, taxa such as Fusobacteriota, Firmicutes, Actinobacteria, and Cyanobacteria were notably less abundant in nos2-/- zebrafish, with no significant alteration observed in monoallelic mutants (Figure 3A)

Line 421- 425: The observed gut microbiota alterations in iNOS-deficient zebrafish, particularly the decline in alpha diversity and microbial community shifts, highlight the essential role of nitric oxide (NO) in gut microbiota regulation. Similar to mammalian models under inflammatory stress, we noted a decrease in Proteobacteria and Firmicutes, in line with iNOS's influence on microbial composition.

Line 489- 491: In conclusion, our study successfully established a stable iNOS-deficient zebrafish model, which preserved normal physiological functions yet exhibited altered nitric oxide production and gut microbiota composition
